# Aging with Autism Departs Greatly from Typical Aging

**DOI:** 10.3390/s20020572

**Published:** 2020-01-20

**Authors:** Elizabeth B. Torres, Carla Caballero, Sejal Mistry

**Affiliations:** 1Psychology Department Center for Biomedicine Imaging and Modelling, CS Department and Rutgers Center for Cognitive Science, Rutgers University, Camden, NJ 08854, USA; 2Sports Science Department, Miguel Hernandez University of Elche, 03202 Alicante, Spain; ccaballero@umh.es; 3Biomathematics Department, Rutgers University, Camden, NJ 08854, USA; Sejal.Mistry@hsc.utah.edu

**Keywords:** stochastic analyses, classification methods, probability distance

## Abstract

Autism has been largely portrayed as a psychiatric and childhood disorder. However, autism is a lifelong neurological condition that evolves over time through highly heterogeneous trajectories. These trends have not been studied in relation to normative aging trajectories, so we know very little about aging with autism. One aspect that seems to develop differently is the sense of movement, inclusive of sensory kinesthetic-reafference emerging from continuously sensed self-generated motions. These include involuntary micro-motions eluding observation, yet routinely obtainable in fMRI studies to rid images of motor artifacts. Open-access repositories offer thousands of imaging records, covering 5–65 years of age for both neurotypical and autistic individuals to ascertain the trajectories of involuntary motions. Here we introduce new computational techniques that automatically stratify different age groups in autism according to probability distance in different representational spaces. Further, we show that autistic cross-sectional population trajectories in probability space fundamentally differ from those of neurotypical controls and that after 40 years of age, there is an inflection point in autism, signaling a monotonically increasing difference away from age-matched normative involuntary motion signatures. Our work offers new age-appropriate stochastic analyses amenable to redefine basic research and provide dynamic diagnoses as the person’s nervous systems age.

## 1. Introduction

Typical neurodevelopment is highly dynamic and variable, with transient changes that tend to occur asynchronously across a given age group. Starting at early infancy, some babies may grow very fast and develop mature neuromotor control before others of the same age group. Other babies may in contrast, be stunted in their growth and/or neuromotor control development [1]. These disparate processes span non-uniform trajectories that vary highly from person to person, across the general population. Despite these known features, developmental research tends to assume an overall consistency of the developmental paths characterizing aging trajectories, whereby linear, parametric models use normal distributions and enforce precisely expected developmental milestones. When faced with a neurological condition like Autism Spectrum Disorders (ASD), the high heterogeneity of any given random draw of this population, then poses a challenge to science, since the assumptions of normality and linearity fail in ways that we do not fully understand.

Open access data can provide information to address these questions on neurodevelopment, so elusive today to basic science. In autism, we know very little about the structure of the probability distribution landscape spanned by the parameters that we can derive from the nervous systems’ self-generated biorhythms. There is a paucity of work with limited sample size addressing the stochastic shifts and maturation trends across the population that is aging with autism [2,3,4]. Consequently, we have virtually no knowledge about typical trends in the maturation of the nervous systems in large cross-sections of the aging population; let alone on the population aging with autism spectrum disorders (ASD.) Analytical tools in ASD research are largely dominated by a “one size fits all” static model that does not yet consider age-appropriate shifts in probability spaces derived from neuromotor variability.

There is an increasing need to stratify these heterogeneous data generated by clinical diagnoses rooted in observation of behavior [3]. Some efforts have been recently appreciated in the literature [5,6,7,8,9]. Nevertheless, the recent efforts have yet to consider age-appropriate shifts in statistical landscape and provide a framework that encourages doing so in relation to normative data, also absent from all clinical diagnostics tools that inform and largely steer today basic research in autism [10]. The exclusion of the motor axis from the current inventories by the Diagnostics Statistical Manual DSM-5 from the American Psychiatric Association APA [11] and by other tools from psychology (e.g., ADOS [10,12]) makes it difficult to use movements and their kinesthetic sensations as aids to objectively categorize different forms of autism as they evolve with aging. Large open access repositories offer a new avenue to complement clinical criteria with physical biometrics derived from variations in human biorhythms from the nervous systems.

In recent years, the micro-movements spikes (MMS) fluctuations inherently present in and underlying all natural behaviors that clinicians observe and describe using pencil and paper means, have been assessed across a proposed taxonomy of neuromotor control [13] (Figure 1). This phylogenetically orderly taxonomy proposes different levels of maturation in neuromotor control under typical neurodevelopment. Because e.g., autonomic function matures earlier than voluntary control, and likewise, reflexes evolve in early infancy, etc. the model also proposes that studying autism across these levels may help us stratify the heterogeneity of autism across aging. Over years of peer reviewed work, using this model has served to reveal several stochastic features of voluntary [2,14], spontaneous [15], involuntary [15,16,17,18] and autonomic [19,20] motions, offering classification power in autism and other neuropsychiatric and neurological conditions (e.g., schizophrenia [21], Parkinson’s disease [22,23,24,25], neuropathies [26] and impairments of the nervous systems due to traumatic brain injury inducing coma [27] or stroke [28]).

In autism, voluntary MMS have provided evidence for a maturational process absent in the autistic phenotype and served as aid to (cross-sectionally) classify different developmental stages according to levels of motor noise evolution and probability distribution shape well-correlated with measures of IQ [2,4]. Since we always sample neurotypical controls and characterize their signatures first, we can always go back to this taxonomy and measure the departure of autistics from the neurotypical signatures. In this paper, we focus on the involuntary levels of neuromotor control, because involuntary motions contribute to the peripheral feedback that the central nervous system continuously senses as the person behaves. When in excess, or when random and noisy, this source of self-generated neurofeedback may interfere with overall neuromotor control and autonomy of the brain over the body in motion. Yet, because these micro-motions are so subtle, they may scape the naked eye of the observant clinician. We have previously found excess motor noise in autism, but have yet to characterize it as a function of aging [16].

Indeed, this recent body of work points at the use of MMS criteria as a potential tool to track the evolution of some of the biorhythms of the nervous systems as the neurotypical person ages. We could then use the normative data to build a proper notion of stochastic change, in a probabilistic sense. This would enable us to better understand the aging process of the autistic nervous systems in cross-sections of the aging population, beyond developmental periods and define age-dependent biometrics capturing different rates of change in young highly plastic systems vs. in aging systems whereby neurodegenerative processes manifest. A pressing question in any lifelong neurological condition like ASD is, when does the plasticity of a young system phase-transitions into a neurodegenerative process? Forecasting this transition would allow us to provide neuroprotective therapies in the rapidly emerging ASD aging population.

In this paper, we present new analytical methods to automatically, in a heuristics-free manner, stratify patterns of variability inherently present in the involuntary head motions of neurotypical controls and of participants with ASD. Their involuntary micro-movements spikes data are reconstructed from brain images, registered as they try to volitionally control their body to prevent it from moving. As they deliberately attempt to dampen movements’ fluctuations upon being instructed to remain still in the fMRI setting, their involuntary motion data reveal the probability landscape of undesirable motor noise across sections of the ASD developing children and the ASD aging population. We explore data drawn at random from over 2144 records of the ABIDE repository and report on fundamental departures of the person aging with autism from the typically aging person.

## 2. Materials and Methods

### 2.1. Demographics and Boot Strapping Technique to Form Age Groups of Equal Size

Across all sites of ABIDE, informed consent was obtained from all participants and/or their legal guardians. The following method have been published previously in various papers [16,17,18], but we provide them here as well to facilitate the flow of the paper. All datasets included in this study are from the Autism Brain Imaging Data Exchange (ABIDE) databases:

ABIDE I (http://fcon_1000.projects.nitrc.org/indi/abide/abide_I.html) and

ABIDE II (http://fcon_1000.projects.nitrc.org/indi/abide/abide_II.html).

ABIDE obeys the following guideline on the use of human subject’s data: “In accordance with HIPAA guidelines and 1000 Functional Connectomes Project/INDI protocols, all datasets have been anonymized, with no protected health information included.”

The study includes two main comparisons:

Autism Spectrum Disorder (ASD), and Typical Development (TD), using estimation of stochastic signatures of involuntary head micro-movements of individuals with a formal DSM-ASD [11] diagnosis of ASD and TD controls.

Ranges of age. Each group (ASD and TD) was split in seven different groups according to their age to assess how the stochastic signature of involuntary head micro-movements evolves with aging. The ranges of age used to that end were the following: from 5 to 10 years old, from 11 to 15 years old, from 16 to 20 years old, from 21 to 25 years old, from 26 to 30 years old, from 31 to 40 years old and from 41 to 65 years old.

Inclusion/Exclusion Criteria

This study includes all sites publicly available through ABIDE I and ABIDE II. They were comprised of 1127 TD and 1017 ASD. As we explained above, those groups were divided by age. Table 1 provides the number of participants with ASD or TD are in each range of age in ABIDE dataset.

#### Bootstrapping Method

ABIDE has data deposited from various labs whereby two types of data are available: cleaned up data sets (whereby noisy images have been removed according to some threshold of head motion amplitude) and raw data sets (whereby no images have been removed). In this work we use the raw data sets with no removal, because we are precisely interested in the head motion data. To that end, we aim at examining all frames continuously acquired in each scan session. Further, we note that different scanner sampling rates influence the nature of the noise in the images, which in turn is reflected in the noise derived from the variability of the raw extracted from the images reflecting the patterns of head motions. Our lab previously reported these effects on the linear (mm/s) and angular (rad/s) speed data [18]. Here we focus on the standardized MMS waveform (see explanation below). We analyze the patters of variability in standardized deviations of the raw signal relative to the empirically estimated mean amplitude, considering randomization across all possible samples. To that end, we use bootstrapping and randomize across our original sample sets.

Furthermore, given the inconsistent group sizes per age groups extracted from the ABIDE datasets (see Table 1), we used a bootstrapping method to ensure uniform group numbers for pairwise statistical comparisons across ages. To that end, we used random sampling with replacement and created subgroups drawn from the original size group while considering the minimum number n = 25 at a time. The time series waveforms from these 25 randomly chosen participants were pooled into a representative point of a given age group. We chose 25 because the smallest age’s sub-groups size was n = 30. Thus, after dividing the groups by age, we did extract 100 and 500 random representative points (as described) to form sub-groups of 100 and 500 group sizes from all the age’s sub-groups. The five-year increments were motivated by prior work highlighting medication use and clinical characterizations of this ABIDE ASD cohort. The standardized amplitude using the MMS waveform ensured shuffling of data across all types of sampling resolution in ABIDE and accounted for anatomical disparities across the different age groups [30]. In this paper we focus on the 500 size groups, but consistent trends were found with 100 size groups.

### 2.2. Data Processing

#### 2.2.1. Motion Extraction

Head motion patterns were extracted from imaging data during (rs) fMRI experiments. Motion extraction was performed using the Analysis of Functional NeuroImages (AFNI) software packages [31]. Single-subject processing scripts were generated using the afni_proc.py interface [31]. Skull stripping was performed on anatomical data and functional EPI data were co-registered to anatomical images. The median was used as the EPI base in alignment. Motion parameters: three translational (*x*, *y*, and *z*) and three rotational (pitch-about the x axis, roll-about the y axis, and yaw- about the *z* axis), from EPI time-series registration was saved.

#### 2.2.2. Statistical Analyses

We assess the scan-by-scan speed-dependent variations in the linear displacement of the head during resting-state functional magnetic resonance imaging (rs-fMRI) sessions. The analyses specifically refer to the stochastic signatures of micro-movement spikes (the MMS, defined in prior peer reviewed work [2,16,17,18]) and their treatment as a continuous random process, under the general rubric of Poison random processes. Specifically, we use a Gamma process and estimate the continuous family of probability distributions using maximum likelihood estimation (MLE) with 95% confidence.

#### 2.2.3. Micro-Movement Spikes Data Type

The maximum amplitude of the speed (mm/s) was obtained (Figure 2A shows the raw waveforms and Figure 2B the MMS, which is the normalized version of the raw peaks). The empirically estimated mean speed of each trial was also obtained and used as reference to determine the maximal absolute amplitude deviations from it.

The time-series of these fluctuations in maximal amplitude deviations from the empirically estimated mean provides the waveform of interest for our analyses. To remove allometric effects of body-size across ages in each trial we computed the normalized peak amplitude (the peak speed amplitude is divided by the sum of the peak speed amplitude and the averaged speed amplitude value comprising points between the two speed minima surrounding the local speed peak amplitude) [32,33]. These normalized peaks spanning real values in the [0,1] interval are the spike trains of random fluctuations in signal amplitude (speed in this case) (Figure 2C). They are assumed to characterize a continuous random process, where events in the past may (or may not) accumulate evidence towards the prediction of future events. The normalized fluctuations define the MMS of the waveform [2] (Figure 2C).

### 2.3. Distance in Probability Space: The Earth Mover’s Distance

The Earth Mover’s Distance (EMD) [34,35], also known as the Kantarovich-Wasserstein distance [36], measures the distance between two discrete probability distributions. Given two discrete distributions *P_r_* and *P_θ_*_,_ each with m possible states x or y, respectively; the EMD computes how much mass needs to be moved how far, to turn one distribution into the other. Figure 2D shows two sample distributions derived from the linear speed that we obtained from head involuntary motions of two participants, one TD and one ASD, while Figure 3A shows the matrix of pairwise EMD quantities across all 500 TD participants in the 5–10-year-old group.

The problem of finding the amount of work that it takes to turn one distribution into another has been posed as an optimization problem that can be resolved using linear programming [35,37] in O(n^2^log(n)) vs. a more recent version in O(n) [38]. Briefly, if one considers these histograms as two piles of earth, there are infinite ways to move the earth from one pile to the other. The goal is to find the optimal one (a unique value despite non-unique ways to do it). These are known as transport problems, with transport plan *γ*(*x,y*), to distribute the amount of earth from one place x over the domain of y (or vice versa.)

Here the constraints $\underset{x}{\sum }\gamma \left(x,y\right)=P_{r}\left(y\right)$ and $\underset{x}{\sum }\gamma \left(x,y\right)=P_{\theta }\left(y\right)\;$ apply to ensure that the plan yields the correct distributions (sum over rows and columns respectively). The *γ* is the joint probability distribution such that $\gamma \in \left(P_{r},P_{\theta }\right)$ and $\Pi \left(P_{r},P_{\theta }\right)$ is the set of all distributions whose marginals are *P_r_* and *P_θ_* respectively. To obtain the EMD, every value of the matrix γ is multiplied with the Euclidean distance between x and y:(1)$$EMD\left(P_{r},P_{\theta }\right)=\underset{\gamma \in \Pi }{inf}{\left\langle D,\Gamma \right\rangle }_{F}$$
where ${\left\langle ,\right\rangle }_{F}$ is the Frobenius inner product (the sum of all the element-wise products) and inf is the infimum (minimizing the expression on the right). This represents an expectation with respect to the joint distribution γ where the outcome is unique but the flow from the minimization process is not unique. The product can be physically interpreted as mass times distance, (work), so Equation (1) is thought of as minimizing the work to transform one distribution into another [35].

We use the EMD to compute pairwise for each age group of each cohort, the cost of turning one probability distribution into the other.

### 2.4. Stochastic Analyses

The foundation of these methods has been explained elsewhere (e.g., [2,3,16,17,18,39] among others). The empirically estimated mean was obtained using the continuous Gamma family of probability distributions for every group because it gave the best fit according to maximum likelihood estimation, MLE). Then, the MMS served as input to the Gamma process and a stochastic characterization of their fluctuations in amplitude was used to characterize the signature of involuntary head motions in the ASD vs. typically developing TD groups.

We examined the frequency histograms of the MMS waveform and use MLE to approximate the best fitting distribution encompassing all cases. To that end, we compared different families of probability distributions (e.g., the Gaussian, Normal, Lognormal, Exponential and Gamma) and chose the best fit in an MLE sense. Owing to our prior works using the ABIDE sets [16,17] we could determine that the Gamma had the best fit in an MLE sense. As such, we settled on the continuous Gamma family of probability distributions [40]. The estimated parameters were plotted on a Gamma parameter plane, where the *x*-axis represents the shape parameter value and the *y*-axis represents the scale parameter value. Figure 2E shows the representation of the estimated Gamma parameters (shape and scale) on the Gamma parameter plane with 95% confidence intervals. Figure 2F shows the empirically estimated probability density functions (PDFs), while Figure 2G shows their representation as points on a Gamma moments multidimensional parameter space. There, the mean, variance and skewness are represented as three dimensional points along the x, y, z axes, respectively. The fourth moment, kurtosis, is proportional to the size of the marker (higher kurtosis is represented by larger marker size) and the color conveys other information. For example, here we use the color to represent the participant type (ASD vs. TD) and the arrow marks the shift in ASD signature relative to the normative value. The color can also provide information on pairwise distance in some other space. We examined the cross-sectional trajectories representing the evolution of these parameters using canonical Cartesian coordinates and Euclidean distance to quantify change and their rates, along with the cumulative distances traveled by each cohort, and other kinematic parameters that such a representation of the probability data affords us. We also represented the EMD as explained above.

### 2.5. Network Connectivity Analyses

The MMS derived from the linear and angular speed were used to group the normalized peak amplitudes into a frequency histogram for each participant. Then, the EMD [34,35,37] was obtained pairwise across all 500 records to estimate the cost of transforming one probability distribution into another. An adjacency matrix derived from this pairwise distance quantities (Figure 3A) was used to obtain an undirected weighted network representation of each group, where the EMD was the weight of the links at entry (*i,j*) connecting two records from participants *i* and *j*. Common network connectivity metrics (e.g., the clustering coefficient, the network’s node-to-node distances and the characteristic pathlength) were used to characterize the neurotypical cohort spanning the 7 age groups with similar number of participants (500 selected at random through bootstrapping, see above) and to provide the normative states of the network. Then, network states generated from the data of the ASD cohort were compared against the normative states.

The distributions of EMD values was also examined (Figure 3B) to ascertain if the values spanned a multimodal distribution or a unimodal one. The Hartigan’s dip test of unimodality was used to that end [41]. Then, the count per bin and the edges information was used to plot the count curve (as in Figure 3C). The curve thus obtained was smoothed using loess algorithm set to 0.1 value using the smooth function in MATLAB version R2018b. The loess algorithm was used to preserve the peaks and valleys. Other smoothing procedures failed to preserve the original local minima and local maxima locations of the original histogram. To automatically separate self-emerging clusters of values using the local minima and local maxima spanned by the smooth count curve, the *findpeaks* function in MATLAB was used. Figure 3D, E show one example for the group of neurotypical controls between 31–40 years of age where the procedure was applied to select the peaks and valleys. The peak amplitudes (local maxima) were located and the values of the EMD that they comprised obtained (see arrows delimiting the clusters bounded by local minima values surrounding the local maximum value.) Then all local maxima and minima were pooled, to determine the full range of values across all age groups and both cohorts. This set of EMD values from the entire data set (7000 points from 500 participants in each of the 7 groups of the neurotypical controls and of the ASD) was sorted and used to build a color code for each participant. To that end, each *i*-participant’s EMD value from the *i,j* matrix entry was mapped (as in Figure 3D–E) to the corresponding distribution bump and the corresponding local maxima of the bump was used to color the point representing the participant in the scatter of probability points plotted on the Gamma moments parameter space. Figure 3F locates on the normalized color scale the value corresponding to the local EMD maximum for participant *i* (matrix entry *i,j*) taken across all other participants. Then, the point located on the Gamma moments parameter space is colored using this scale.

The idea was to uncover self-emerging clusters of EMD values in each age-group. Low values represent distributions that are similar, while high values represent distributions that are far apart in probability space. Possible scenarios are to have within each age group, highly heterogeneous values, or highly homogeneous values. Further, across the full scatter representing all age groups, we may have highly heterogeneous values, or homogeneous values. We set to interrogate this probability landscape about the evolution of the age groups, the heterogeneity/homogeneity of the groups and the nature of the cross-sectional trajectories of the groups.

To interrogate about trajectories, we used the mean values of the three first Gamma moments in each cohort and traced a line connecting each representative point of each age group for each cohort. We also did this for the mode of the age-group scatter, representing the most frequent value and for the median (not shown). The coordinates of the mean values along each axis, were set as the representative point of the scatter and tracked over time. The departure of the coordinates of the point representing the ASD age-group from the neurotypical control age group were obtained using the norm of the vector difference and plotted to ascertain the trends of the trajectories and to identify inflection points of the evolving curve representing the point by point distance between the two cohorts. Lastly, the trajectories were plotted in the scatter of points representing the probability landscape to ascertain their locations within the scatters.

A measure of the scatter dispersion was built using the Delaunay Triangulation (DT) of each age-group scatter and the area of each of the triangles obtained. The fluctuations in the values of the area were treated as a point process and their MMS obtained as well. The values of the DT-area MMS were used to empirically fit the Gamma distribution (given that this was the best distribution using MLE [21]) and the estimated shape and scale parameters were plotted on the Gamma parameter plane with 95% confidence intervals for the empirically estimated shape and scale parameters. We used the estimated PDFs to ascertain differences of statistical significance across all age groups of the cohorts and within each age group for the ASD and neurotypical cases.

## 3. Results

### 3.1. High Heterogeneity of ASD is Automatically Captured by the Earth Movers’ Distance

The stochastic analyses empirically estimating the moments of the continuous family of Gamma probability distributions, derived from the linear speed MMS, provided a way to represent the estimated stochastic signatures in a parameter space. In this space, each point represents a participant, and the scatter is plotted using canonical Cartesian coordinates. They fall at a distance from each other in this space that we can measure through e.g., Euclidean distance. Yet, since these scatters of points are probability functions, an appropriate measure to examine their similarity is the Earth Mover’s Distance (EMD, see methods). Here we use the distribution spanned by the EMD from participant *i* to all other participants and index the maximal value representing the farthest apart a person is from all other members of the age group, to color code the point in the scatter. Doing this through the full cohort, automatically identifies self-emerging clusters of participants, indicating important differences within a group and across age groups. Cross-sectionally, these patterns inform us about the aging trajectories of ASD in relation to the age-matched normative data and in so doing the empirically estimated personalized signatures of involuntary micro-motions automatically stratify ASD by age (Figure 4).

Figure 4 shows the fundamental differences between the normative age-groups and those of the ASD age-matched groups. In the neurotypical control group, the EMD shows a more homogeneous quantity in each age group, with higher ranges of EMD. This suggests that each cluster is uniquely positioned in probability space far apart from other clusters. It also indicates that each participant within each age cluster, is unambiguously different from all other participants. In contrast to the normative data, the ASD groups also colored by the EMD, is rather heterogeneous. There, the points in probability space are more mixed, some closer to others and some farther apart. And this is a visible pattern up to 20 years of age.

A bird’s view of this cohort confirms the notion that ASD is highly variable. But there is another feature that this parameter space and coloring reveals. There is a cluster that completely separates from the rest of the scatter (the oldest group of 41–65 years of age). This indicates a fundamental shift in the signatures of involuntary head micro-motions after 40 years of age. Interestingly, after 20 years of age, the spread in the ASD scatter of each age group decreases, and so does the value of the EMD. The latter, indicating that the participants are getting closer in probability space. This trend is maintained right until 31–40 years of age, when the inflection point marks a fundamental departure in the stochastic signatures of involuntary head motion variations. It is right at this point that the EMD increases maximally as the scatter spread contracts. All participants in this 41–65-year-old group become unambiguously different (far apart) in probability space and rather far from all other age-clusters in the Gamma moments parameter space.

The breakdown of these scatters by age are shown in Figure 5 (for the TD cohort) and Figure 6 (for the ASD cohort). There, it is possible to appreciate the above-mentioned shifts of the points across the Gamma moments parameter space, for each age group in the typical cohort of Figure 5. Further, we see the homogeneity and low EMD values of the 31–40-year-old ASD group. Not only they fall tightly close on the Gamma moments parameter space, they are also close in probability space, according to the EMD values. The evolution of the stochastic signatures on the Gamma moments parameter space and those of the EMD in probability space denote a fundamentally different cross-sectional trajectory of the aging process in ASD, relative to the normative data. The oldest age group (41–65 years old) completely separates from the rest of the cohort and this dramatic transition in the involuntary head motions can be forecasted before 40 years of age.

### 3.2. Differences in the Scatters Representing Each Age Group

To ascertain the shifting differences in the spread of the scatter of points representing the empirically estimated Gamma moments of the MMS derived from the involuntary linear speed of each participant, we obtained the Delaunay triangulation of the scatter. For each unique triangle found by the Delaunay algorithm, we computed the area and studied the patterns of variability of the triangles’ areas across each age group within each cohort.

Figure 7 shows that for each age group, the ASD scatter had a fundamentally different spread signature than the neurotypical scatter. This is shown through the empirically estimated PDF representing the signatures derived from the Delaunay triangulation that we obtained from each group’s scatter. The square symbol localizing the shape and scale of the PDF on the Gamma parameter plane is for ASD participants. The circle symbol is for the age-matched TD participants. Further, the figure reflects the shifting nature of the scatter spread variability. In each age group, the empirically estimated Gamma PDF characterizing the spread of the ASD scatter shifted with respect to the age-matched controls, as shown by the PDF insets of each corresponding Gamma parameter plane. The color range represents the average Earth Mover’s Distance (normalized between 0 and 1) between the PDFs representing the variability of the triangles’ areas in the Delaunay triangulation of the scatter. For example, in the 41–65 group, the yellow circle representing the controls has a triangulation with lower scatter variability than the red square representing the ASD participants of that age group. The shift in PDF is evident in the differences in shape and scale (dispersion) of the empirical distributions.

### 3.3. Aging with ASD Is Fundamentally Different from Typically Aging

Tracking the cross-sectional trajectory of the estimated mean value of each age-group scatter, as one moves from age group to age group, revealed a very different cross-sectional aging path in neurotypical controls than in ASD participants. This can be appreciated in Figure 8A, where we plot the ASD trajectory in red and the age-matching controls in blue, connecting across each estimated mean along each of the dimensions. There, we can see that ASD starts different from the age matched controls at 5–10 years old. More importantly, we see that the ASD groups evolve at a much lower rate, thus falling behind in the trajectory, as the controls advance to the point representing the 11–15-year-old group. This slow trend remains, so moving onward to the point representing the 16–20-year-old group, upon which they accelerate.

The Euclidean distance cumulatively traveled in this Gamma parameter space by the ASD groups comprising 5–20 years of age is ½ of that traveled by their age-matched control counterparts. This can also be appreciated in the acceleration and deceleration of the curves in Figure 8B and in the shifts of the linear increments along the positional path. This trajectory culminates with a large separation of the oldest ASD group from the age-matched controls, visibly in Figure 8A and also captured by the pairwise Euclidean distance plot in Figure 8D. Figure 8E,F show the scatters of the TD and ASD respectively with the trajectories superimposed and shown in a rotated view that helps visualize the fundamental departure of the oldest group (after 41 years of age) from the rest of the population in ASD.

### 3.4. Network Connectivity Metrics Derived from the Autistic Cohort Reveal Fundamental Departures from Normative Data

Comparison of the autistic cohort to the normative data revealed statistically significant differences in the network connectivity metrics derived from the EMD adjacency matrix (see methods). Each entry in this matrix quantified the pairwise distance in probability space between two members of the age group within each cohort. Figure 9A,B illustrate an example of the resulting patterns from the comparisons using the clustering coefficient, a metric that ascertains the degree to which the nodes of the graph tend to cluster together. Higher values indicate higher clustering. From this figure it is evident that the younger ASD groups (5–20 years of age) are significantly more heterogenous than their age-matched controls (*p* << 0.001, Figure 9C), with several clusters within each age group within those first 20 years of life.

Each age group within the typical and autistic cohorts had statistically different clustering coefficient values according to the non-parametric Kruskal-Wallis test. Likewise, pairwise comparison between each neurotypical and autistic age-group yielded statistically significant differences according to this non-parametric ANOVA test (Figure 9C) (*p* << 0.001).

Another network metric, the characteristic pathlength quantifying the average short-distance paths between a node and all other nodes of the network, was also obtained for each age-group in each cohort (Figure 9D). Large differences were found in the autistic cohort relative to the normative data. The groups with minimum characteristic pathlength (5–10 years old) and that with the maximum characteristic pathlength (31–40 years old) were also quantified. They coincided in both the neurotypical and the autistic cohorts, with large differences of the autistic groups relative to the controls. Figure 9E,F show the representative adjacency matrices for these groups along with the matrix quantifying the pairwise network distances for each age group within each cohort. Here we see that at maximum characteristic pathlength the adjacency matrix has low values indicating pairwise more similarity in the probability distributions of the involuntary micro-movement spikes. This is the case for both controls and ASD, occurring in both cases for the 31–40-year-old groups. The entries in the network distance matrix indicate the length of the shortest distance path between the two nodes (*i, j*) of the matrix entry. This number is computed based on the travel weight of the network edges, which in our case represent the EMD. The length thus denotes how similar (or different) the family of probability distributions are to get from participant *i* to participant *j*. Long travelled distances represent paths along edges that are maximally apart in probability space. At the minimum characteristic pathlength occurring for the 5–10-year-old group, the EDM matrix is much higher for ASD, thus suggesting higher heterogeneity than the age-matched controls. The network distance matrices however have low values, so the shortest pairwise paths are comparable for both groups. Owing to the heterogeneity in probability space, for any two participants within the age group and cohort (ASD or control), it is always possible to find shortest travel path hoping along points of low value in EMD (i.e., points that are close in probability space.).

In the case of maximum characteristic pathlength, occurring for the age group of 31–40 years old in both the neurotypical control and the ASD cohort, there is appreciably less heterogeneity of shortest distance values in ASD than in the age-matched controls, and the values are significantly lower than those of controls (*p*<<0.001, rank sum test.) Here a more homogeneous ASD group gives rise to lower travel pathlength on average than those of the distribution points from age-match controls. The ASD lower heterogeneity of this age group in probability space appears to be forecasting a dramatic change in the involuntary movements of ASD after 40 years of age (as suggested by Figure 6).

## 4. Discussion

This work aimed at using a distribution-free approach to characterize the cross-sectional aging trajectories of the stochastic signatures of involuntary micro-movements in neurotypicals from 5–65 years of age. The work further aimed at quantifying the departure from the normative data of age-matched participants with ASD. To that end, instead of applying a grand average method which would eliminate as noise the important motor fluctuations from the involuntary motions, we examined the MMS in two different spaces. One space was the canonical representation of points located on a Gamma moments parameter space, where differences are measured using Euclidean distance. The other was the probability space, where we use the EMD to measure similarities/differences between points. Each point represents the probability distribution of a person’s MMS derived from involuntary linear displacements of the head, registered as they try to remain still. We used a Gamma process to characterize the spike data. Together, at each age group, these points represent a family of continuous probability distribution functions. This family was well fit by the continuous Gamma family using Maximum Likelihood Estimation (MLE) with 95% confidence [2].

We found that each age group could be well characterized by a different family of probability distributions, with different scatter dispersion. The methods provide a new way to automatically stratify the autistic cohort using the EMD on the probability distributions generated by the MMS, and to characterize the differences from the neurotypical cohort by representing the data as weighted connected graphs. The network analyses provided several connectivity metrics amenable to automatically separate atypical autistic development from neurotypical development. More importantly, we were able to automatically identify critical differences in the aging trajectories of the autistic cohort that were absent in the neurotypical cohort for the eldest group (41–65 years of age).

These results are relevant to autism research in more than one way. They emphasize the importance of not enforcing a priori any statistical assumptions, but rather allowing the variability inherently present in the data to automatically reveal self-emerging patterns. Further, the results show that a “one size fits all” approach is inefficient to cope with the heterogeneity of autism. Each age group in the autistic cohort had a different stochastic signature unique to the group, spanning a family of PDFs. These were distinguishable from other age groups within the ASD cohort. This was also the case for the normative data, which was nonetheless, overall, visibly more homogeneous in nature than the ASD set. At an individual level, each person in each age-group of the normative cohort was unambiguously statistically different from every other person. This was captured by the large values of the EMD, signaling high effort to transform one frequency histogram into any other frequency histogram of the cohort.

In stark contrast to the normative trends, the ASD group had non-uniform changes in the first 20 years of life. On the Gamma moments parameter space (Figure 8A) a representative 11–15-year-old ASD child has stochastic signatures closer to a representative 5–10-year-old neurotypical child, and then accelerates to the 16–20 range, which remains behind normative values. The irregular trends in the acceleration and deceleration period separating the cohorts are visible in Figure 8B,C across ages. As the two cross-sectional cohorts age, these differences increase, suggesting fundamental neurological disparity in the involuntary motions. These results are congruent with prior cross-sectional data from goal-directed voluntary motions, where the dramatically different trajectory in the maturation of the somatic-sensory-motor systems had been revealed and are well characterized by a power law [2,3] and by a parameter space defined by the R-metric [4]. Here, using new methods, we extend the results from voluntary to involuntary motions across a much larger range of ages and larger cohorts that open-access data can afford us.

Beyond the MMS related work in ASD, the broader literature of motor control problems studying movements and their sensations in ASD has by now grown considerably [42,43], including problems with gait disturbances and vestibular dysfunction from an early age [44,45]. The current clinical diagnostics criteria do not include these problems as core symptoms. Yet, for well over two decades now, basic science has provided objective, physical quantification of neurological problems in ASD that supports an earlier neurological model [46,47] amenable to explain many of the social and communication issues defining the condition. Patterns of cerebellar dysfunction [48,49], motor and vestibular coordination [44] and sensory-motor integration issues [2] have been reproduced across tractable biorhythms using non-invasive means [50,51,52,53,54,55,56]. Perhaps these data can help us derive further criteria to stratify ASD and help design treatments tailored to address the phenotypic features that each subgroup may reveal. More importantly, using the neurological model in combination with the current behavioral descriptors may be effective to tackle these somatic sensory motor issues from an early age.

Under the current behavioral criteria for diagnosis and treatments of ASD, these neurological issues are treated as comorbid and secondary to the observed behaviors defining the condition. Under such criteria, the adult autistic population, which is rather large today, never received neuroprotective therapies aimed to slow down the progression of neurological issues. Yet, Parkinsonism is more prevalent and has earlier onset in autistic adults than in neurotypicals of comparable age [57]. Other medical conditions are also more prevalent in autistic adults than in controls of similar age [58]. According to a recent report by US-Kaiser, medical issues include all major psychiatric disorders such as depression, anxiety, bipolar disorder, obsessive-compulsive disorder, schizophrenia, and suicide attempts. Further the report cites immune conditions, gastrointestinal and sleep disorders, seizure, obesity, dyslipidemia, hypertension, and diabetes with higher prevalence in adults with autism, along with stroke and Parkinson′s disease.

The results presented here strongly suggest that the involuntary micro-movements can not only provide a way to characterize the lack of volitional control of the brain over the body (i.e., control the body at will to remain still). More importantly, the results offer for the first time the means to differentiate aging with autism from typically aging in this cross section of the population. The methods allowed us to automatically stratify the broad spectrum of autism as a lifelong condition. The cross-sectional aging trajectories of these stochastic signatures pinpoint critical points of change in the statistical landscape of development and aging. These may perhaps forecast important milestones signaling physiological changes in the autistic system.

Along those lines, a striking feature of the data concerns the separation of the 41–65-year-old cluster from the rest of the scatter. It may be possible that the high penetrance of Fragile X Syndrome in autism accounts for such differentiation, as motor symptoms are common among this part of the population and often result in Parkinsonism after 40 years of age [59,60,61]. It will be important to address this question, as the dramatic change in the involuntary motions for this ASD age group raises concerns about the large number of aging adults with ASD who may face motor issues impeding independent living.

### Limitations

We note that among the autistic participants of ABIDE, there are those who have taken psychotropic medications and those who have not. We have previously shown that the involuntary MMS are present in both types of participants, but that as the number of meds increases, so does the noise levels and the randomness of these involuntary motions [4,16]. This is relevant to these types of analyses, as they are very sensitive to medication effects. Further, we have shown that these MMS can differentiate those individuals with ASD who never took meds from those do did and identify patterns of medication intake across ages in the ABIDE repository. Likewise, the MMS have differentiated ASD with and without comorbidities (e.g., ADHD) and identified levels of severity according to clinical scales commonly used for research (e.g., ADOS and IQ test scores reported on ABIDE.) Although these may be possible advantages of the methods and MMS data type, we note that one limitation is the novelty of the approach, as very few labs have used these methods. Nevertheless, our lab is making them available through various means to invite reproducibility of results across open access databases such as ABIDE, Kaggle.com, the connectome project and others.

Along those lines, the sensory-motor axes [62] were only recently added on January of 2019, to the Research Domain Criteria (RDoC) matrix created by the National Institute of Mental Health (NIMH) [63]. As such, the sensory motor issues discussed here have not been broadly explored in autism, although there is a nascent field researching the neurological issues underlying behaviors across the spectrum of autism (e.g., [42,46,49,50,51,54,56] among others) is beginning to take advantage of the wearable sensors revolution and measure the biorhythms of the peripheral nervous systems signals [2,15]. This is important as this form of feedback, which is disrupted in autism, is critical to continuously guide the central controllers of the brain generating observable behaviors that define autism today. In this sense, the present results warrant further research in autism across the lifespan. This type of work may mark the start of a new medical era for ASD, a prospect that to some may appear as a limitation, while to others, may seem advantageous.

## 5. Conclusions

It is hard to imagine a point whereby a neurodevelopmental condition may turn into a neurodegenerative disorder of the nervous systems, particularly since autism is primarily conceived as a mental illness, a cognitive/communicative childhood disorder and hardly defined and treated as a neurological condition. Perhaps it may be time to rethink autism as a lifelong condition that requires close attention to the physiological changes that may accelerate the process of neurodegeneration in the early adult or elderly years. This paper opens a new conversation regarding autism motor physiology across the human lifespan.

## 6. Patents

EBT holds the US Patent “Methods and Systems for the Diagnoses and Treatments of Nervous Systems Disorders” combined in the paper as micro-movement spikes, MMS data type and Gamma process.

## Figures and Tables

**Figure 1 sensors-20-00572-f001:**
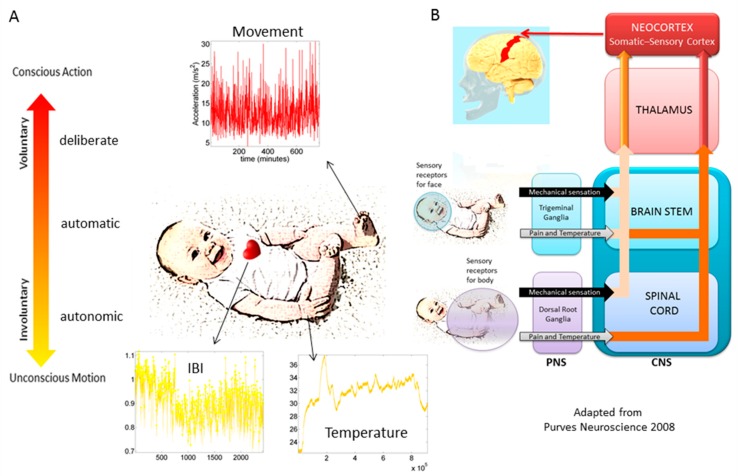
Proposed taxonomy of neuromotor control with phylogenetic order of maturation to stratify autism spectrum disorders (figure reproduced from Chapter 1 in [29], with permission from Elsevier). (**A**) Current wearable biosensors enable non-invasive co-registration of multiple biorhythms streamed from different levels of somatic-sensory-motor control, from the face and body. (**B**) Fluctuations in these biorhythms from the different levels of the taxonomy are modeled as standardized unitless micro-movement spikes MMS. It is proposed that these continuous spikes serve as a form of kinesthetic reafference, mapping the sensory consequences of actions according to different levels of neuromotor control (e.g., autonomic sensory consequences vs. voluntary sensory consequences). These in turn, map differently for trigeminal and dorsal-root ganglia systems defining socio-motor axes that could be used to stratify autism’s social differences (e.g., eye control, auditory, communication, taste, smell and swallowing issues in the face, vs. pointing, balance, gait, coordination, vestibular issues of the body -as possible scenario whereby data is readily available using non-invasive, off-the-shelf means to categorize autism.

**Figure 2 sensors-20-00572-f002:**
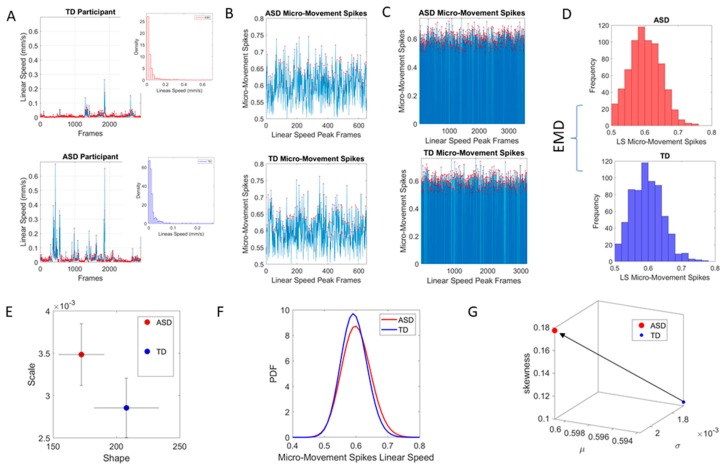
Methods: (**A**) Raw data in the form of linear speed (mm/s) extracted from images of ABIDE using traditional methods to clean the motor artifacts generated by involuntary motions of the head, as the participant attempts to remain still upon instruction during resting state fMRI. Peaks denoting fluctuations in amplitude are highlighted in red dots. Insets show histograms of the raw peaks. (**B**) The micro-movement spikes (MMS see methods explanation) are extracted from the moment by moment fluctuations in peak amplitude (see methods for normalization). They are unitless spike trains and the normalized peaks are gathered in frequency histograms (see insets in A) (**C**) Full micro-movement spikes from all original frames are used to ascertain statistical differences using the same number of frames in the original raw data. (**D**) The frequency histograms of the peaks are compared using the earth mover’s distance (EMD see methods explanation) quantifying the cost to transform one histogram into the other. (**E**) Comparison of stochastic signatures on the Gamma parameter plane spanned by the shape of the empirically estimated distribution and the scale (noise to signal ratio) indicating the dispersion of the PDFs in (**F**). (**G**) Gamma moments are used to build a scatter of points on a parameter space where the first three moments (mean, variance and skewness) are represented as x, y, z axes and the fourth moment, the kurtosis is reflected in the size of the marker. A fifth dimension is used to represent the color of the marker using e.g., the range of EMD values determined in (**D**).

**Figure 3 sensors-20-00572-f003:**
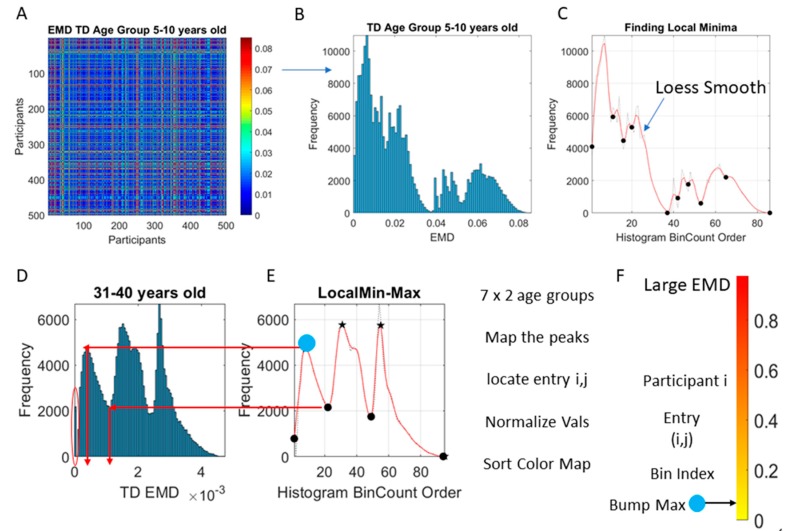
Methods: (**A**) Sample adjacency matrix built using the pairwise EMD across all 500 participants of the 5–10-year-old group. (**B**) Multi-modal frequency histogram of EMD values across the matrix entries. (**C**) Curve of the bin count order represented in the jagged dashed curve smoothed using loess method to localize the local minima and maxima. (**D**) Another sample multi-modal histogram and multi-peaked curve derived from it to explain the process of clustering. Each local minima and maxima are automatically detected along the histogram bin count order spanning the red curve and mapped back to the original histogram of EMD values. As such, given a participant i vs. all other participants, we can localize its maximal distance across the entire age group as the blue circle in (**E**). (**F**) These values are then mapped to a normalized scale and used to color the individual points in the scatter according to their values. For example, here the blue dot representing the local maxima from the first bump is localized along the color bar representing the full range spanned by all age groups in the TD and ASD cohorts.

**Figure 4 sensors-20-00572-f004:**
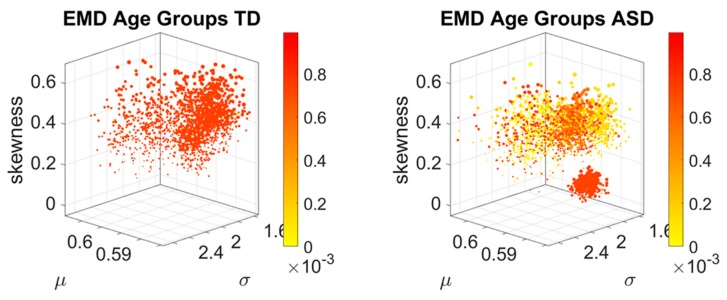
Normative stochastic signatures of cross-section of neurotypical population vs. Autism Spectrum Disorders (ASD) signatures from age-matched groups, empirically estimated from the involuntary head displacements (micro-movement spikes reflecting fluctuations in linear speed). Scatters show high heterogeneity in ASD according to the ranges of pairwise Earth Mover’s Distance (EMD) values (color coded normalized while considering the full range of values from both cohorts.).

**Figure 5 sensors-20-00572-f005:**
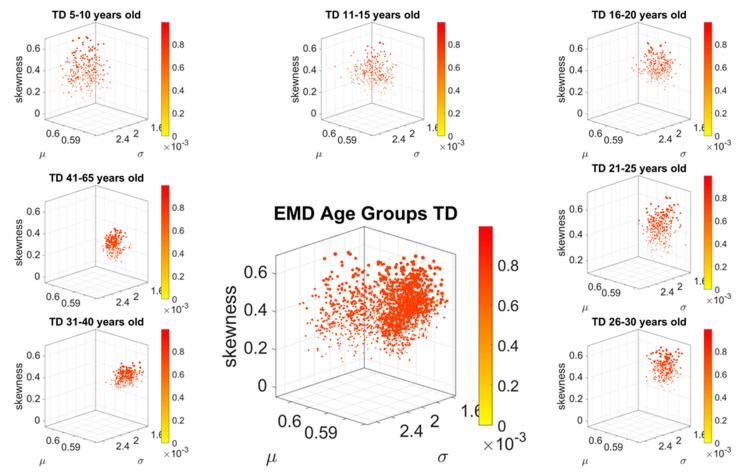
Smooth stochastic shifts of Gamma moment parameters characterize typical aging, with broader spread in the scatter for younger years tending to decrease the spread with aging. Pairwise ranges of EMD values are high, denoting unambiguous maximal separation from each member of the age group to every other member. The normalized color map uses the full range of neurotypical controls and ASD values across all age groups in both cohorts.

**Figure 6 sensors-20-00572-f006:**
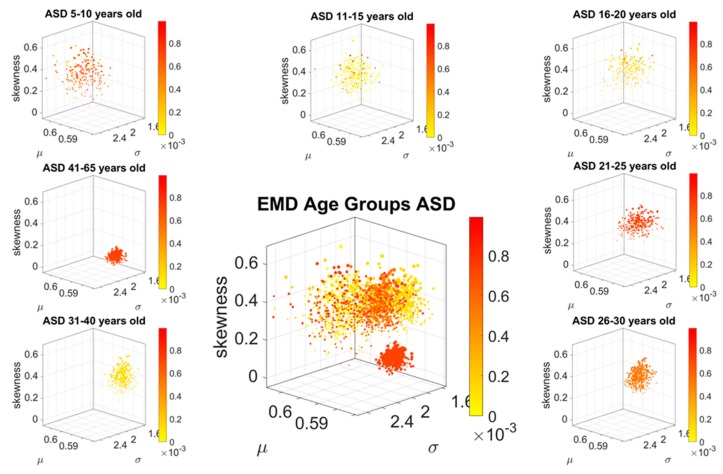
Stratification of ASD by EMD capturing the pairwise differences in probability distributions empirically estimated from the MMS representing the involuntary fluctuations in the linear displacements of the head. Notice that each scatter is different in spread and range of EMD. Earlier years up to 20 are more heterogeneous than later years in that they have a broader spread in the Gamma moment parameter space that is accompanied by heterogeneous distances in probability space. This trend changes with aging. After 20 years of age, the scatter spread decreases and the EMD maximally separating each individual within the group from the rest of the age group becomes more homogeneous. Each person’s signature is close in probability space. Their fluctuations in involuntary motions become statistically closer. Then, by 40 years of age they are maximally close in probability space and their scatter’s spread tends to shrink. After 40 years of age, there is a fundamental departure of the ASD signatures from the rest of the cohort. Compare these scatters to those in Figure 2 representing the normative data, to best appreciate that aging with ASD is different from typical aging. The trajectory evolution of the stochastic signatures in the probability landscape serves as a natural classifier of autism’s involuntary motions.

**Figure 7 sensors-20-00572-f007:**
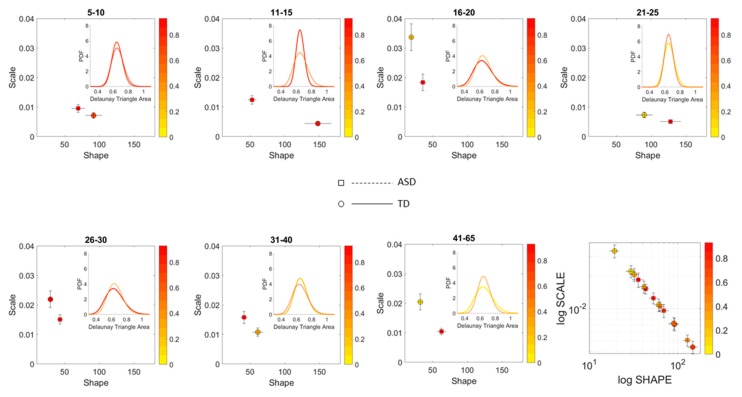
Characterization of the scatters’ spread using the patterns of variability of the area of the triangles making up the Delaunay triangulation in each age group. Color map represents the average EMD of the triangle area PDFs of each cluster taken as a normalized scale across the full range of values in both TD and ASD scatter spreads. Each three-dimensional scatter in Figure 5 and Figure 6 representing the stochastic signatures of the involuntary MMS from the linear speed was characterized by a different Delaunay triangulation and the triangles’ areas spanned probability distribution capturing the variability of the scatter. Each age group spanned a statistically separable scatter when comparing normative data vs. ASD, empirically fitted by different probability distribution (dashed PDFs are from the ASD scatters). In the last right most panel, the evolution of the stochastic signatures of involuntary movements’ variability over the 5–65 years of age period can be appreciated as a trajectory on the log-log Gamma plane parameter space.

**Figure 8 sensors-20-00572-f008:**
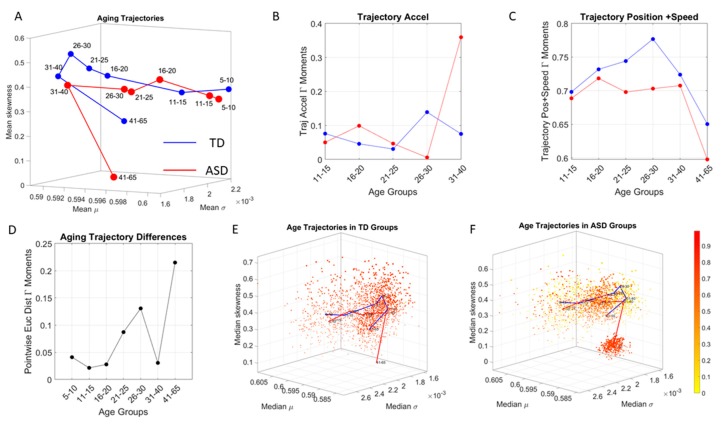
Aging trajectories captured cross-sectionally in each cohort (**A**) Dots denote the location of the mean values of the empirically estimated Gamma moments of each age-group and group type. These show different trajectories with non-uniform changes in ASD during the early years and large departure from normative data from age-match neurotypically developing controls (TD) throughout aging, particularly after 41 years of age. Notice that the rates of change of the shifts in the points are different in ASD. They do not change as regularly as the controls during the first 15 years. (**B**) The acceleration of the trajectory shows the differences in the rates of change of the shifts in the averaged Gamma moments for each cohort, with inflection points around 26–30 years of age that take the systems into different directions. (**C**) The position plus each increment (trajectory velocity) is also accumulated to examine the trajectory over time. This reveals differences that become more pronounced after 20 years of age and decrease by 40, right before the largest departure from normative data that separates the oldest ASD cohort form the age-matched TD. (**D**) The pointwise difference between the two trajectories reveals a large change after 20 and an inflection point by 40, right before the accelerated change of the 41–65 ASD group. (**E**) The trajectories are embedded in the scatter to give a sense of the differences in relation to the full TD cloud. (**F**) The cloud of ASD with the embedded trajectories highlights the large departure of the 41–65 group from the normative trajectory.

**Figure 9 sensors-20-00572-f009:**
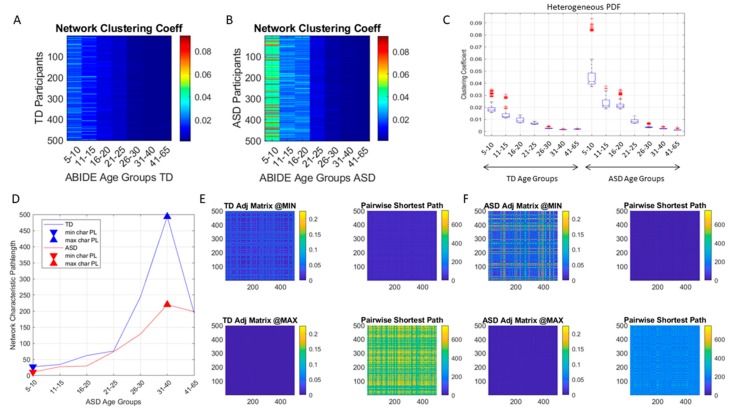
Network connectivity metrics used to separate the two cohorts by age groups and automatically detect significant statistical differences across age groups and between each control and ASD same-age group. (**A**) Values of network’s clustering coefficient for each node (representing a participant) in neurotypical controls distinguish each age group. (**B**) Likewise, the clustering coefficient distinguishes each age group, with higher values in ASD relative to controls. Nodes are more connected in earlier years (up to 20 years of age) in ASD according to the similarity metric of EMD computed pairwise and determining the weighted connected graph of each cohort. (**C**) Non-parametric ANOVA, Kruskal-Wallis test captures the within and between group differences. (**D**) The characteristic pathlength computing the average shortest distance path from all nodes to each node also captures differences within each group and between the two cohorts of typical controls and ASD. Minimum and maximum characteristic pathlength values are shown with corresponding adjacency matrices (**E**) and network distance matrices (**F**) for each cohort.

**Table 1 sensors-20-00572-t001:** Number of subjects for each age group extracted from ABIDE.

AGE GROUP	ASD	TD
**5 TO 10 YEARS OLD**	228	265
**11 TO 15 YEARS OLD**	374	417
**16 TO 20 YEARS OLD**	200	178
**21 TO 25 YEARS OLD**	103	116
**26 TO 30 YEARS OLD**	43	76
**31 TO 40 YEARS OLD**	39	43
**41 TO 65 YEARS OLD**	30	32

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
