# Peer review of "Aging with Autism Departs Greatly from Typical Aging"

_sensors, 2020, doi:10.3390/s20020572_

Round 1

Reviewer 1 Report

Manuscript Number: sensors-676498

Title: Aging with Autism Departs Greatly from Typical Aging

General Comments:

The current study addressed an important issue on the neurodevelopmental landscape of autism. The authors used a proposed taxonomy of neuromotor control to help us stratify the heterogeneity of autism across aging. The current project primarily focused on the involuntary levels of neuromotor control and use a new analysis to automatically stratify patterns of variability on the involuntary head motions of individuals with and without ASD. The idea is novel and the analysis is sophisticated. The findings provide important information on the “motor” characteristics of autism population. It would be even better if the authors could relate these motor measures with clinical severity of ASD. Because I am not familiar with the analysis in the current paper, I will leave other reviewers to provide comments on the methods. The following are my detailed comments for clarification.

Introduction:

Page 3, line 87, the first paragraph: could you briefly review how this model has been used to classify autism. It would be very helpful to provide some background on why you think the involuntary movements can be an “window” to understand the developmental landscape of autism. The review of literature is not so clear here. Could you provide any hypotheses on both early development and aging?

Methods:

I am not an expertise on this type of analysis. I will leave other reviewers for commenting this part.

Results:

Figure 7: The label should be in color as well (i.e., ASD in orange and TD in red)?

Discussion:

It is interesting that the most significant ASD group is the eldest group (41-65 years of age). For the rest of age groups, due to the individual differences for TD, the significance of ASD can be occluded. Now, could you please address the clinical implication of these findings? How these findings explain/address the current diagnose criteria of ASD? Could you address any limitations of current analyses? If IQ or medication could impact the key measures, have you considered to use partial correlation or other methods to minimize the effects?

Author Response

Please see the comments in the attached word doc and also on the new updated version of the MS that addresses the important feedback of the reviewer.

Reviewer 2 Report

The paper is overall well written and has been pleasant to read. Material and methods are very dense and extensive but I consider the degree of statistical depth necessary to justify the results. The research question is interesting and the conclusion: “Autism as a lifelong condition that requires close attention to the physiological changes that may accelerate the process of neurodegeneration in the early elderly years”, should be considered as very important for Scientifics and Clinicians. The methodology is novel, “Large open access repositories offer a new avenue to complement clinical criteria with physical biometrics derived from variations in human biorhythms from the nervous systems” and considered as a complement of clinical criteria. “The results show that a “one size fits all” approach is inefficient to cope with the heterogeneity of autism”.

I would like to highlight that “the results offer for the first time the means to differentiate aging with autism from typically aging in this cross section of the population” and that “using the neurological model in combination with the current behavioral descriptors may be effective to tackle these somatic sensory motor issues from an early age”. The manuscript brings us closer to the question for any lifelong neurological condition like Autism: when does the plasticity of a young system phase-transitions into a neurodegenerative process?.

I propose these minor changes:

Line 122: change “shave” for “have”

Line 108: remove “These participants range across age groups of 5-65 years of age.” It is specified in material and method section.

Line 329: Abbreviations must be specified in the text in the figure 4, Earth Movers’ Distance (EMD) and autism spectrum disorders (ASD)

Line 355: the same than figure 4 for abbreviations

Line 379: the same than figure 4 for abbreviations

Lines 553-561: This paragraph should be re-written as limitations, starting with a sentence including “Limitations”. Appendix A (not appendix figure 1) should be removed because it is not used as limitation.

Author Response

We have addressed the reviewer's comments in the attached word doc and pointed the reviewer to the locations in the MS where we changed the text according to the reviewer's suggestion.

We sincerely thank the reviewer for the constructive comments and hope that our edits add clarity and improve the MS.

Reviewer 3 Report

I think the manuscript includes novel and intriguing fondings.

It is thus almost acceptable for publication in the present form.

However, the authors should revise it according to the following minor 

concerns.

The authors should describe on the reason why the micro-movements spikes(MMS) fluctuations of head motion patterns were selected  in the present study.  As shown in the manuscript title, the authors have demonstrated that ageing with ASD departs greatly from typical aging. The authors should discuss more in detail on the underlying mechanism of the above findings, citing relevant literatures.

Author Response

We thank the reviewer for the constructive feedback and have addressed the comments with the hopes of gaining clarity and improving the MS.
